# Influence of Silver Nanoparticles, Laser Light and Electromagnetic Stimulation of Seeds on Germination Rate and Photosynthetic Parameters in Pumpkin (*Cucurbita pepo* L.) Leaves

Agata Dziwulska-Hunek [1] , Magdalena Kachel [2] , Mariusz Gagoś [3] and Mariusz Szymanek [4,*]

1   Department of Biophysics, University of Life Sciences in Lublin, Akademicka 13, 20-950 Lublin, Poland; agata.dziwulska-hunek@up.lublin.pl

2   Department of Machinery Exploitation and Management of Production Processes, University of Life Sciences in Lublin, Głęboka 28, 20-612 Lublin, Poland; magdalena.kachel@up.lublin.pl

3   Department of Cell Biology, Maria Curie Skłodowska University, Akademicka 19, 20-031 Lublin, Poland; mariusz.gagos@poczta.umcs.lublin.pl

4   Department of Agricultural, Forest and Transport Machinery, University of Life Sciences in Lublin, Głęboka 28, 20-612 Lublin, Poland

*   Correspondence: mariusz.szymanek@up.lublin.pl

**Abstract:** The study aimed to determine the impact of laser light (L), magnetic stimulation (*p*) and silver nanocolloid (AgNC) on pumpkin seeds on the germination rate and content of photosynthetic pigments as well as the efficiency of photosynthesis and greenness index in the plant's development stages. Seeds germinated after the use of various combinations of different refining techniques. The best results were observed for the alternating magnetic field, where the germination energy increased significantly by 20% relative to the control. A similar effect was observed in terms of germination capacity which increased by 4%. A decrease in terms of emergence rate was observed in all study groups. Leaves grown from seeds soaked in nanocolloidal silver on platters were characterised by a significantly higher content of chlorophyll *a* and *b* by, respectively, 53 and 11%, as well as 79% higher carotenoid content. The leaves of potted plants contained 42 and 43% more chlorophyll *a* in groups *p* and AgNC. In addition, 66 and 81% more carotenoids in groups L and *p*. At the stage of the onset of flowering of pumpkin, an improvement in terms of photosynthetic efficiency and greenness index was observed in all study groups. The highest improvement was recorded for seeds soaked in silver and reached 23% (intensity of photosynthesis) and 11% (greenness index SPAD).

**Keywords:** seed; pumpkin; germination; photosynthesis; laser light; silver nanocolloid; magnetic stimulation

## 1. Introduction

Pumpkin (*Cucurbita pepo* L.) is a member of the *Cucurbitaceae* family. It is cultivated in many parts of the world. It is nutritious and offers a range of health and beauty benefits, for instance due to its content of carotenoids which are known for their anti-cancer and anti-oxidative qualities [1–3]. Cucurbita seeds contain approximately 27% protein and 54% fat, which makes them a valuable source of nutrients for both humans and animals [4,5]. They are also rich in vitamins, mineral salts, and pectin [6,7]. The edible parts of Cucurbita include its: pulp, seeds, leaves, and flowers [1]. It finds many culinary applications, particularly as an ingredient of soups, salads, sauces, cakes and desserts [8]. A very important yielding factor is the adequate preparation of seed lot, whose aim is to increase the germination capacity of seeds and then improve the vigor of seedlings [9]. In recent years, the intensification of crop production in farm practices began to actively introduce electrophysical methods of affecting plants and seeds of cereals, vegetables, and legumes [10].

Pre-sowing laser and magnetic bio-stimulation of seeds beneficial effect on germination, seedling growth, and yield have attracted the attention agriculturists [11,12]. A lot of studies which have been conducted so far showed a positive influence of a pre-sowing treatment of seeds with physical stimulants (laser light, magnetic and electric fields, gamma radiation, etc.) on the stimulation of germination, initial development and yielding of selected cereals [11–16]. Reports available in literature also mention increased content of photosynthetic pigments [17–21] and increased protein concentration [17,20] in crops and vegetables grown from seeds subjected to electromagnetic stimulation prior to sowing.

Laser activation of plants results in an increase of their bioenergetic potential, leading to higher activation at fitochrome, fitohormone and fermentative systems, as a stimulation of their biochemical and physiological processes [21]. A number of the most recent studies have also turned their attention to nanotechnological techniques offering many potential applications in scientific, technological, and agricultural contexts [22–25] due to their antibacterial and antifungal benefits [26]. Nanoparticles are now being used in the manufacture household goods, food packaging, feed production, as well as textile and cosmetic products [27–29]. The renewed interest in nanoparticles and nanomaterials stems from their unique and improved qualities related to their size, distribution, and morphological structure. A growing number of researchers are looking into their potential due to the observed increasing microbiological resistance to metal ions, antibiotics, and development of immune bacterial strains [30]. In recent years, numerous studies in the field of agriculture have considered the use of silver nanoparticles (AgNPs) as possible pre-sowing treatment or nano-fertilizer. The researchers considered the effects of silver nano-particles on the height of the developing plant [31], amount of the harvestable biomass [32,33], germination capacity [34,35], seed harvest, and fruit quality [33–36]. The studied uses of silver nano-particles in plant production have not explicitly confirmed the intensification or inhibition of plant growth as a result of the same; instead, it has been observed that the dosage of AgNP to considerable extent determines its harmful or beneficial effects [37,38]. The potential applicability of silver nanoparticles (AgNPs) in agriculture remains a matter of some debate [39,40]. The influence of AgNPs seems to depend on the species and age of the given plant, the size and concentrations of the nanoparticles themselves, as well as the experimental conditions, including: temperature and duration as well as the method of applying the preparation. It is generally accepted that the impact of nanoparticles on plant growth and development can be both positive and negative [41,42]. AgNPs increased the growth rate and biochemical characteristics (chlorophyll, carbohydrate and protein content, anti-oxidative enzymes) of *Brassica juncea*, common bean, and corn [43,44]. Krishnaraj et al. [41] studied the influence of biologically synthesized AgNPs in a hydroponic cultivation on the increase of *Bacopa monnieri* and discovered that biosynthesised AgNPs have a significant impact on the germination capacity of seeds. Furthermore, AgNPs contributed to the intensification of seed germination and seedling of *Boswellia ovaliofoliolata* trees [45]. Crop, vegetable, and leguminous plantations around the world are affected by a variety of stress conditions (draught, water deficits, salination, etc.) which can significantly influence seed germination [46,47]. To answer the same, constant efforts are being made with the view of finding new methods of improving the seed material [48]. Among these, nanotechnology remains a relative novelty. We still do not know the full extent of its potential side effects, i.e., the related mechanisms of toxicity or the consequences of releasing the material into the ecosystem [49]. The study described below focused on verifying various techniques with the potential of improving the germination and emergence rate of pumpkin seeds, determining the respective content of chlorophylls and carotenoids, and photochemical intensity.

The aim of the study was to analyze the effects of seed stimulation using laser light, electromagnetic fields and silver nanocolloid on the germination, emergence rate, content of photosynthetic pigments, and photosynthetic efficiency in the respective stages of the plants' development.

## 2. Material and Methods

### 2.1. Plant Material and Seed Treatment

In the experiment, non-encapsulated seeds of Polish pumpkin cultivar 'Miranda' (Torseed, Toruń, Poland) were used. A pumpkin cultivar that does not have a hard cover (seeds without shell); therefore, its cultivation is not easy. Pumpkin tolerates drought badly, it is sensitive to frosts, both in spring and autumn. The seeds are characterized by a high oil content, so they are sometimes used in extruders. Improving the quality of seed material and, as a result, increasing yields is the most important task of the Polish farm industry.

The seeds were stimulated prior to sowing using the following methods: He-Ne laser light (Figure 1) at the wavelength of 632.8 nm; surface power density of 6 mW·cm$^{-2}$, with the exposure time of 1 min (L); alternating magnetic field with 30 mT induction with the exposure time of 1 min (*p*); silver nanoparticles (AgNPs)—by way of soaking the seeds in silver nanocolloid (AgNC) and non-stimulated samples C (control).

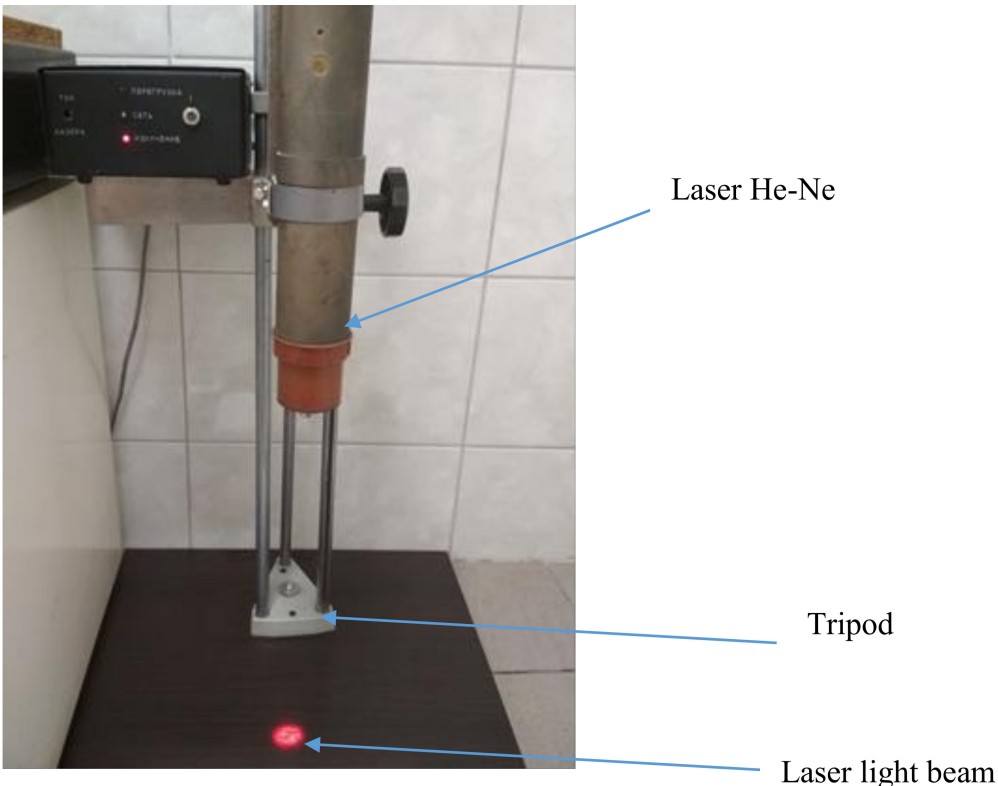

Laser He-Ne

Tripod

Laser light beam

**Figure 1.** Self-designed rig for subjecting seeds to laser light stimulation.

The surface power density was measured with the use of a power meter laser (Model CTL-2001, LaserInstruments, Warszawa, Poland). Stimulation using alternating magnetic field was provided using Petruszewski's magnet [50]. Its design and principle of operation has been described in the relevant paper by Muszyński et al. [51].

The nanomaterial used for laboratory tests was obtained in the form of commercially available nanocolloid concentrated at ≥0.1% silver (ITP-1KAg PO) (1000 ppm AgNC) (by ITP-SYSTEM Sp. z o.o. in Dąbrowa Górnicza, Poland).

Measured proportions of nanocolloid were added in addition to 1 L of sterile distilled water and stirred at room temperature for 15 min 2 mL of the pre-prepared solution of the AgNC concentrated at 50 mL·L$^{-1}$ was poured over groups of seeds intended for the AgNC bath, which were subsequently left to soak for 1 h at the temperature of 20 °C. After one hour, the seeds were strained and sown.

### 2.2. Atomic Force Microscopy (AFM) Analysis

The analysis entailed confirmation of the nanoparticle content in the preparations.

It was performed with the use of an atomic force microscope NanoScope V (Veeco, New York, NY, USA), which allows magnified and three-dimensional imaging of solid surfaces with the magnification rage from 2000 to 500,000 times. The measurement accuracy was consistent with the PN-EN ISO/IEC 17,025 standard. The AFM preparation was obtained by placing a drop of the solution into a mica plate glued to a metal disc and evaporating it. Measurements were conducted at room temperature [52]. The obtained AFM images were processed using nanoscope analysis software (Bruker, Billerica, MA, USA).

### 2.3. The Experiment of Pumpkin Germination

The experiment entailing pumpkin germination on laboratory Petri dishes and in pots in the greenhouse (Table 1). The mean temperature inside the greenhouse was 19 °C in May, 25 °C in June, and 35 °C in July, at the relative humidity of 40%.

**Table 1.** Time-frames the laboratory and greenhouse experiments and BBCH scale.

| Experiment | | Date | BBCH Scale |
|---|---|---|---|
| **Sowinig** | | 13 May 2017 | - |
| **Laboratory (Petri dishes)** | $GE_4$ | 17 May 2017 | - |
| | $GC_8$ | 21 May 2017 | - |
| **Sowing** | | 16 May 2017 | - |
| **Greenhouse (pots)** | $N_{g8}$ | 24 May 2017 | - |
| | $N_{g14}$ | 30 May 2017 | - |
| **Stage of development** | SI | 23 June 2017 | 31–39 |
| | SII | 26 June 2017 | 41–49 |
| | SIII | 23 July 2017 | 51–69 |
| | SIV | 2 August 2017 | 71–79 |

SI—budding, SII—onset of flowering, SIII—flowering, SIV—fructification.

The experimental germination in Petri dishes took place at room temperature of $20 \pm 1$ °C, with 12/12 h insolation. Seeds were sown of 25 per onto Petri dishes (with dimensions of $120 \times 15$ mm) lined with three layers of blotting paper moistened with distilled water (5 mL) in each sample. In the greenhouse, seeds were sown of 6 per pot containing universal soil with the addition of humidifier and NPK fertilizer, in four replicates per each study group. Each sample included four replications. The pots were 25 cm in diameter and 7.5 liters in capacity.

Energy (*GE*) and capacity (*GC*) germination in laboratory tests were calculated from the formulas:

$$GE = \frac{N_{g4}}{N} \times 100\% \qquad (1)$$

$$GC = \frac{N_{g8}}{N} \times 100\% \qquad (2)$$

where:

$N_{g4, g8}$—number of seeds germination after 4 and 8 days.
$N$—number of seeds sown.

In the greenhouse tests, the number of plants after 8 days (emergence) ($N_{e8}$) and 14 days ($N_{e14}$) was given. The measurements were conducted on 10 randomly selected leaves for each of the study groups, in four various stages of the plants' development. The BBCH scale was used to identify the phenological development stages of plants [53].

### 2.4. Spectrophotometric Methods

Spectrophotometric methods were used to analyze photosynthetic pigments in seed leaves from the dishes (after 8 days) and pumpkin leaves from the pots (after 14 days), collected at the stage of emergence. The analysis entailed isolating chlorophylls and carotenoids from the leaves in acetone containing 0.01% BHT—butylated hydroxytoluene (in darkness) to avoid oxidation. Carry Bio 300 dual beam spectrophotometer (Agilent Technologies, CA, United States) was used to measure UV-Vis spectra. Pigment concentrations were calculated in accordance with the Lichtenhaler and Buschmann [54] procedure. The content of chlorophyll was determined by spectrophotometric method in three replications for each research object.

### 2.5. Photosynthetic Activity of Plants

To estimate the actual photochemical activity of PSII in situ, chlorophyll *a* fluorescence in plants was measured using a pulse amplitude-modulated (PAM) fluorimeter (Mini PAM, Walz, Germany) with a light emitting diode at 650 nm and a standard intensity $0.15 \, \mu mol \, m^{-2} \, s^{-1}$ PAR (photosynthetically active radiation). The device's built-in sensor projects light impulses (triggered on and off at a very high frequency) onto a leaf placed in a metal clasp, and registers the fluorescence signal from excited chlorophyll *a* molecules. The following fluorescence parameters were measured: F-the present fluorescence yield, M-fluorescence after saturation pulse was applied, $Y(II) = (M - F)/M$, PAR-photosynthetically active radiation, ETR-electron transfer rate calculated as: ETR = $Y(II) \times$ PAR $\times$ ETR-factor; i.e., ETR = $Y(II) \times$ PAR $\times 0.5 \times 0.84$. The standard factor 0.84 corresponds to the fraction of incident light absorbed by a leaf [20,55]. The study included measurements of the greenness index (SPAD) with the use of chlorophyll meter (model SPAD-502, Konica Minolta, City, Japan) in which a leaf is placed in a clasp and 2 s later the result appears on the screen and the device is ready to scan another leaf. SPAD greenness index and photosynthesis efficiency using Mini-PAM were determined on 10 randomly selected leaves. Moreover, the measurement of the greenness and efficiency of photosynthesis takes place in four phases of plant development BBCH scale (the scale used in the UE to identify the various stages of plants) (Table 1).

### 2.6. Statistical Analysis

The research data were processed by analysis of variance (ANOVA) using the computer program Statistica. The analysis of variance was preceded by checking whether the results are normally distributed with the Shapiro-Wilk test and the homogeneity of variance with the Levene's test. Arithmetical means and standard deviation of the experimental data were calculated. Significant differences at $p \leq 0.05$ among the data were evaluated according to Tukey's LSD test. Correlation analysis was performed to determine the strength and character of the relationships between variables. The correlation was statistically significant when $p \leq 0.05$.

## 3. Results and Discussion

### 3.1. AFM Analysis of AgNC

The analysis consisted in confirming the content of nanoparticles in the preparations.

In order to assess the surface properties and imaging in nanoscale resolutions, the analyzed samples of AgNC nanocolloids were subjected to a morphological analysis with AFM. Images were taken in the two-and three-dimensional AgNC topography format, as presented in Figure 2. The paraemters adopted during analysis enabled the identification of minimum and maximum height values for the AgNC nanoparticles.

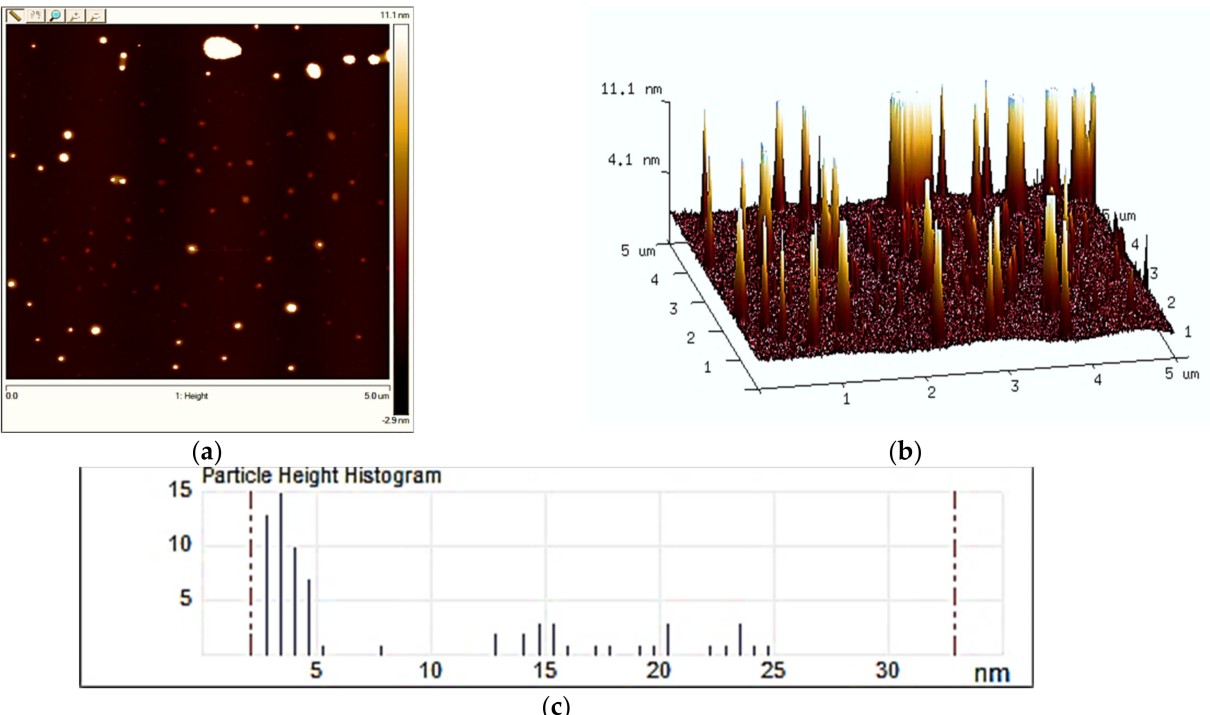

**Figure 2.** Atomic force microscopy results for AgNC: (**a**) 2D image depicting particle height and distribution of nanoparticles per 100 objects, (**b**) 3D image, (**c**) histogram of particle height.

Based on the obtained results (Figure 2), it was concluded that the mean analyzed surface area 16,314.2 nm$^2$, 75 AgNC were observed. The minimum and maximum height of the same was, respectively, 2.136 nm and 32.91 nm.

### 3.2. Germination Seeds

The germination energy was higher in seeds subjected to laser light and magnetic field stimulation and was smaller for silver nanocolloid, relative to the control (Table 2).

**Table 2.** Germination and emergence of pumpkin after applying the respective pre-sowing treatments.

| Parameters | C | L | $p$ | AgNC |
|---|---|---|---|---|
| **Germination** | | | | |
| *GE(%)* | 54 ± 2.31 [(1)] b | 60 ± 13.86 b | 74 ± 12.44 a | 53 ± 16.45 b |
| *GC(%)* | 84 ± 8.64 a | 79 ± 11.02 a | 88 ± 8.64 a | 75 ± 13.61 a |
| **Emergence** | | | | |
| $N_{e8}$ | 6 ± 0.00 a | 5 ± 1.41a | 3 ± 2.12 ac | 1 ± 0.00 bc |
| $N_{e14}$ | 6 ± 0.00 a | 6 ± 0.00a | 3 ± 1.41 b | 2 ± 0.71 b |

[(1)] SD; C—control; L—laser light; $p$—magnetic field; AgNC—silver nanocolloid. Different letters a→c—significant differences ($p \leq 0.05$), the same letters—no difference between the control and study groups.

The conducted analysis showed that a significant increase was observed in relation to the controls for GE by about 37% for $p$. In turn, with regard to $N_{e8}$, a significant decrease was observed for AgNC by about 83%, and, for $N_{e14}$ for $p$ and AgNC by about 50% and 66%, respectively. A significantly positive effect of laser radiation on the germination of seeds was found in white lupine [56], white clover [57], radish [58], cucumber [59] and tomato [60]. No positive effect was observed in oats [61], sugar beet [62], cucumber and tomato [60], and pepper [61]. Similarly to the studies presented here, many authors examined the effect of an He-Ne laser and alternating magnetic field on the germination of amaranth seeds.

Dziwulska-Hunek et al. [63] used the laser beam stimulation of the power density of 6 mW/cm$^2$, the alternating magnetic field with the induction 30 mT, the frequency 50 Hz for t = 30 s as well as the combination of both factors for amaranth seeds cvs Aztek and Rawa. The effects on the germination of the seeds were examined on Petri dishes in darkness at temperatures from 10 to 55 °C. The most pronounced effects important for the germination process were registered at the temperatures of 20 and 25 °C. When the above parameters of the stimulation of amaranth seeds cv. Rawa were present, the increase in dry matter, raw protein, crude fiber, crude ash and in the final yield was observed. An increase in the levels of Leu, Val, Lys, and Phe + Tyr amino acids as well as decreased levels of Arg, Glu, and Ala. The levels of Cys, Tr, Ile, His, and Pro remained unchanged. Sujak and Dziwulska-Hunek [64] applied a laser beam of the power density of 6 mW/cm$^2$, alternating magnetic field with an induction of B = 30 mT and frequency of 50 Hz for t = 30 s, and the combination of these factors to stimulate amaranth seeds. A magnetic field applied to dormant seeds was found to increase the rate of subsequent seedling growth of barley, corn, beans, wheat, certain tree fruits, and other tree species. Moreover, a low frequency magnetic field (16 Hz) can be used as a method of post-harvest seed improvement for different plant species, especially for seeds of temperature sensitive species germinating at low temperatures [65]. Studies conducted by Gubbins et al. [66] and El-Temsah and Joner [67] demonstrated that the use of AgNPs in the concentration of 10 mg·L$^{-1}$ AgNPs inhibited the germinating capacity of *Hordeum vulgare* seeds and reduced the length of sprouts in flax (*Linum usitatissimum*) and barley (*Hordeum vulgare*). Amooaghaiea et al. [68] observed that the use of silver nanocolloid concentrated at 0.2 to 1.6 mg·L$^{-1}$ contributed to the inhibition of seed germination and lipase activity in germinating seeds and seedlings of *Brassica nigra*. In the present study, seeds soaked in silver nanocolloid showed a decrease in germination. Stmapoulis et al. [69] reported that the use of a preparation containing nanoparticles at the size of 100 nm AgNPs at 100 and 500 mg·L$^{-1}$ resulted in a 41 and 57% biomass decrease in a cultivation of *Cucurbita pepo*, relative to the control, whereas, in a study by Abdel-Azeem and Elsayed [70], where silver nanoparticles of varying size were used (65, 50, and 20 nm) to test their impact on the germination capacity of *Vicia faba*, the results clearly indicated that the varying AgNP diameters had only a minor bearing on the process of germination when compared to the control. Similarly, the results obtained by Barren et al. [71] did not demonstrate a significant impact of AgNPs on the germination capacity of *Cucumis sativus* or *Lactuca sativa* seeds. Yasur and Rani [72] also confirmed in their study that the application of AgNC did not have significant influence on the germination capacity of seeds, length of roots, or overall growth of castor bean, *Ricinus communis* L., even at higher concentrations. In a study conducted by Mehrian et al. [73], it was observed that the use of silver nanoparticles in high concentrations of 75 and 100 mg·L$^{-1}$ and sized at 20 nm contributed to a significant increase of stress suffered by plants, and, consequently, to their reduced vitality and lower number of tomato seedlings. In another experiment [23], it was observed that the germination capacity of barley seeds was reduced after the application of AgNPs in various concentrations. On the other hand, results reported by Hojjat and Hojjat [34] suggest that increased salinity of the environment of plant development resulted in reduced germination while application of silver nanoparticles actually improved plant germination parameters.

### 3.3. Elements Concentrations in Leaves of Pumpkin

In the leaves of sprouts grown in Petri dishes from seeds soaked in silver, a significant increase was observed in terms of chlorophyll *a* and *b* content, of respectively: 54% and 11% compared to the control. In all other cases, a decrease in the pigment's content was observed (Figure 3a).

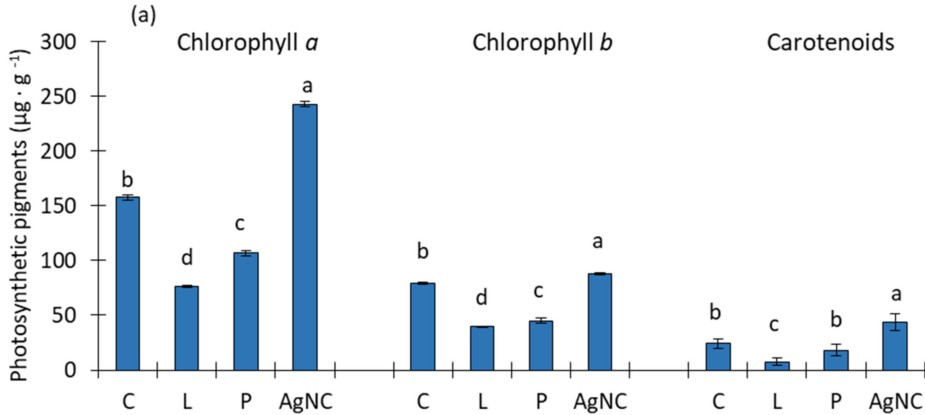

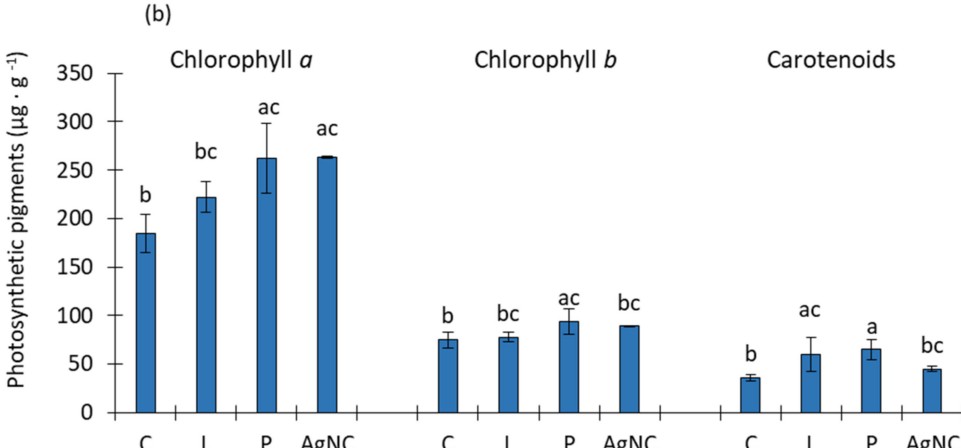

**Figure 3.** Content of photosynthetic pigments in the leaves of pumpkin plants grown from seeds subjected to the respective pre-sowing treatments: (**a**) leaves from dishes, (**b**) leaves from pots. C—control; L—laser light; *p*—magnetic field; AgNC—silver nanocolloid. Different letters a→d—significant differences ($p \leq 0.05$), the same letters—no differences. Statistical analysis performed separately for Chlorophyll *a*, *b*, and carotenoids.

The concentration of carotenoids in the leaves of sprouts grown in the dishes, in the sample treated with AgNC, was significantly higher, by approximately 79%, relative to the control (C). A reduction in the pigment's concentration was observed in samples stimulated with laser light and magnetic field. Leaves from plants grown in pots (Figure 3b) revealed increased chlorophyll *a*, *b* and carotenoid content in all study groups. The highest significant values of chlorophyll *a* were recorded for the *p* and AgNC samples, which amounted to 42 and 43%, respectively, compared to the control. Moreover, a significant content of chlorophyll *b* was found only in trial *p*, where it was 25% more than in relation to C. Stimulation with laser light and alternating magnetic field significantly influenced the content of carotenoids, with an increase of 66 and 81%, respectively. In another study conducted by Liu et al. [74], the use of green LED light resulted in a decrease in chlorophyll and carotenoid content in lettuce. In a study by Liu et al. [74] conducted on cucumber plants, saline stress caused a reduction in photosynthetic efficiency. The content of chlorophyll depended on the development phase and the method used. Nanotechnology can have a beneficial effect on the increase in photosynthetic pigments concentration in chickpea according to the results reported by Anusuya and Banu [49]. In a study by Fayez et al. [22] it was observed that silver nanoparticles applied in various concentrations caused a decrease in chlorophyll *a*, *b* and carotenoid content in the leaves of barley, relative to the control. According to Kachel et al. [75], the content of photosynthetic pigments in the leaves of pumpkin with nano silver (concentration of 50 mL L$^{-1}$) decreased after 22 days, and, after

29 days, there were more of them than in the control group. The increase was 5% (Chl a and Carotenoids) and 10% (Chl b). In our study, at the same concentration of Ag nanoparticles for pumpkin leaves after 8 (Petrry dishes) and 14 days (pots), the concentrations of chlorophyll *a* (43–54%), chlorophyll *b* (11–19%), and carotenoids (25–79%) compared to the control. In other studies on rapeseed oil produced from seeds treated with nanoparticles, regardless of the harvest year or the methods used, the content of chlorophylls and carotenoids was significantly higher than the control samples chlorophyll content in plants was also analysed by Mazumdar [76]. Based on the obtained results, he concluded that silver nanoparticles in *B. campestris* significantly reduced the total chlorophyll content relative to the control when applied in the concentrations of 500 µg·L$^{-1}$ and 1000 µg·L$^{-1}$. Moreover, negative effects were also observed at the dose of 1000 µg·L$^{-1}$ Ag nanoparticles applied in *V. radiata* and *B. campestris* with regard to total chlorophyll content compared to the control. In a study conducted by Asghar et al. [12], the researchers reported increased chlorophyll content in soya plants grown from seeds stimulated with He-Ne laser light and alternating magnetic fields. As observed by Winkelmann et al. [77], silver nanoparticles can interfere with cells' biological functions due to two distinct mechanisms. A nanoparticle may be absorbed by a plant cell and start to release silver cations as products of various redox reactions. Cells can also fully absorb nanoparticles in a process known as endocytosis. It is less common in the case of larger nanoparticles due to their size. Once inside a cell, a nanoparticle may release Ag$^+$ or silver atoms which then bond with proteins and hinder their functional capacity. One of the signs of efficiently working defence mechanisms against oxidative stress is a plant's ability to produce chlorophylls and carotenoids. Chlorophyll particles undergo degradation when a plant is under conditions of abiotic stress caused by absorbing excessive amounts of metals with toxic effects on the plant. In their study, Qian et al. [38] demonstrated that two-week-long exposition of *A. thaliana* cuttings to AgNPs resulted in a decrease in their Chl *a* and Chl *b* content. AgNPs applied in concentrations of 0.5 and 3.0 mg·L$^{-1}$ significantly reduced the plant's Chl *a* content to the level of 60.9% compared to the control in both cases. Moreover, the above concentration of AgNPs (0.5 and 3.0 mg·L$^{-1}$) also triggered a decrease in Chl *b* levels.

The correlations could be observed between the particular groups: C × C (r = −0.360), L × L (r = 0.856), *p* × *p* (r = 0.982), AgNC × AgNC (r = −0.967) for pots and C × C (r = −0.152), L × L (r = 0.971), *p* × *p* (r = 0.353), AgNC × AgNC (r = −0.086) for dishes (Table 3).

**Table 3.** Pearson correlation coefficient between chlorophyll *a* + *b* and carotenoids.

| | | Pearson Coefficient r | | | |
| | Research Groups | Carotenoids | | | |
| | | C | L | *p* | AgNC |
|---|---|---|---|---|---|
| Chlorophyll *a* + *b* (in leaves dishes) | C | −0.360 | 0.603 | −0.983 | 0.382 |
| | L | 0.012 | 0.856 | −0.980 | 0.011 |
| | *p* | 0.365 | −0.599 | 0.982 | −0.3877 |
| | AgNC | 0.960 | 0.269 | 0.449 | −0.967 |
| Chlorophyll *a* + *b* (in leaves pots) | C | −0.152 | 0.749 | −0.985 | −0.916 |
| | L | 0.324 | 0.971 | −0.953 | −0.625 |
| | *p* | −0.943 | −0.791 | −0.353 | −0.223 |
| | AgNC | 0.795 | 0.940 | 0.623 | −0.086 |

Calculations of the correlation coefficients (Table 3) between the values of total chlorophyll and carotenoids revealed no correlation between the Petri dish and pots study groups. As one of the characteristics increased, so did the other, which may suggest that the two are correlated with each other. In the case of seed leaves grown on dishes, no correlation between total chlorophyll *a* + *b* and carotenoids was observed.

Compared to the controls, a reduction of mean content of chlorophyll *a* + *b* in dishes (Figure 4) was observed for L by about 50% and for P by about 35% an increase of AgNC

by about 38%. With regard to the carotenoids, the decrease was for L and P and increase by AgNc by about 68%, 27% and 79% respectively.

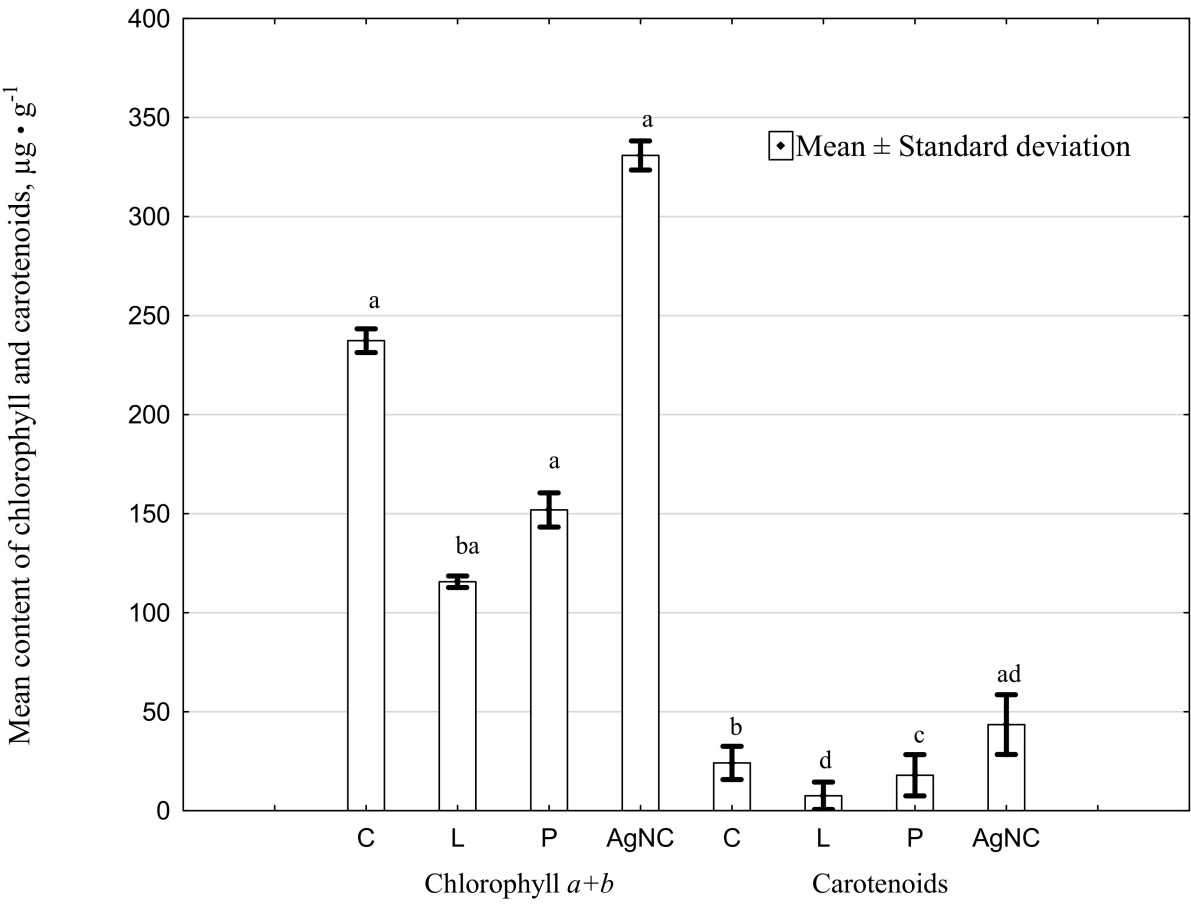

**Figure 4.** Mean content of chlorophyll *a* + *b* and carotenoids (in seed leaves grown in dishes for 8 days). C—control; L—laser light; *p*—magnetic field; AgNC—silver nanocolloid. Different letters a→b—significant differences ($p \leq 0.05$), the same letters—no difference between the control.

Compared to the controls, an increase of mean content of chlorophyll *a* + *b* in pots (Figure 5) was observed for L by about 29%, for P by about 28% and for AgNC by about 47%. With regard to the carotenoids, the increase was observed for L, P and AgNc by about 29%, 55% and 15% respectively.

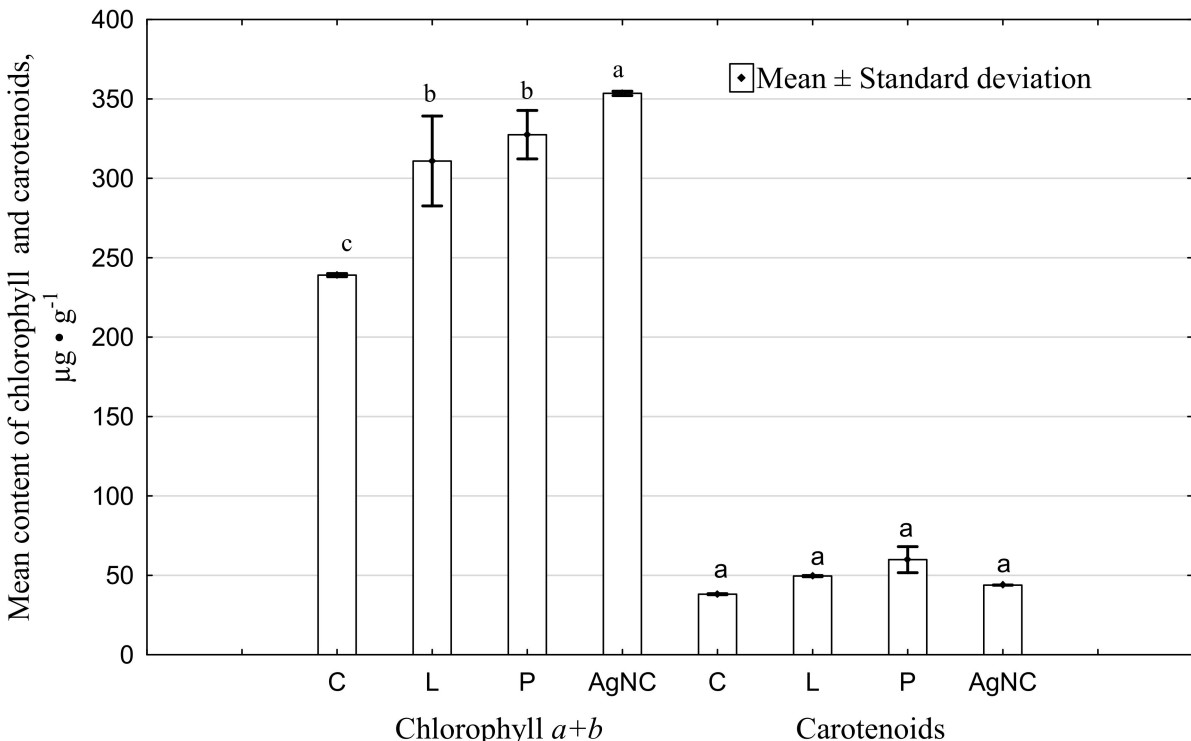

**Figure 5.** Mean content of chlorophyll *a* + *b* and carotenoids (from seed leaves grown in pots for 14 days) in the respective study groups. C—control; L—laser light; *p*—magnetic field; AgNC—silver nanocolloid. Different letters a→b—significant differences ($p \leq 0.05$), the same letters—no difference between the control.

PAR radiation was low (Table 4), ranging from 57 (L, SIII) to 302 μmol of photons m$^{-2}$ s$^{-1}$ (AgNC, SIV). The value of photosynthesis intensity for the low PAR was, respectively, 0.713 and 0.646. Light intensity in all study samples was reduced at the SII and increased at the stage of SIV when compared to the control. Photosynthetic efficiency at the onset of SII was increased in all study groups compared to the control. The highest increase was observed in the group treated with AgNC and reached 29%. Low intensity of PAR means that a large dose of energy was absorbed, which had a positive effect on photosynthesis, which, in plants soaked in silver, reached 0.728. Low PAR intensity resulted in increased photosynthetic activity at SII while reducing the same at the SIV. The described studies indicated an increase in terms of photochemical efficiency and greenness index at the SII in all study groups when compared to the control. The highest values of photochemical efficiency and greenness index were observed in leaves grown from seeds soaked in AgNC. The same were respectively 26 and 11% higher than in the control. According to a study by Kumar et al. [78] on safflower plants (*Carthamus tinctorius* L.) treated with laser beams (wavelength 632.8 nm) at different times (0.5 h, 1 h, 1.5 h) showed a differentiation in photosynthetic activity, but its increase was noticed compared to the control. In our research, laser light had a variable effect on the photosynthesis efficiency in pumpkin leaves depending on the stage of development. It was the highest in the SII (14%) and SIII (4%) phases. In other studies on long-term laser beam irradiation (wavelength 632.8 nm), it was noticed that the photosynthetic activity did not differ significantly between the research objects [79]. According to research study by Li et al. [80] regarding the effect of red (R), white (B) and mixed light (RB) on the photosynthesis efficiency in red pepper leaves, it was noticed that the leaves were thicker and the ability to transpose electrons in the process of photosynthesis and its speed increased by lights B and RB.

**Table 4.** Chlorophyll fluorescence parameters in the leaves of pumpkin grown from seeds subjected to the respective pre-sowing treatments.

| Chlorophyll Fluorescence Parameters | Stage of Development | C | L | $p$ | AgNC |
|---|---|---|---|---|---|
| F | SI | 785 ± 70.45 [1] a | 867 ± 98.30 a | 803 ± 89.18 a | 465 ± 77.69 b |
| | SII | 392 ± 109.97 b | 412 ± 57.10 ad | 432 ± 63.59 ad | 593 ± 104.20 ac |
| | SIII | 565 ± 68.82 a | 573 ± 91.16 a | 517 ± 121.87 ac | 624 ± 101.60 ab |
| | SIV | 632 ± 78.62 b | 732 ± 14.20 ac | 657 ± 111.42 bc | 300 ± 74.63 ad |
| $F_m$ | SI | 2465 ± 458.68 a | 2317 ± 573.89 a | 2694 ± 447.17 a | 1622 ± 213.46 b |
| | SII | 1006 ± 386.54 b | 1241 ± 251.49 b | 1143 ± 217.55 b | 2180 ± 239.68 a |
| | SIII | 1813 ± 257.01 a | 1993 ± 255.61 ab | 1616 ± 441.41 ac | 1901 ± 239.71 ab |
| | SIV | 2149 ± 445.82 a | 2195 ± 423.10 a | 2143 ± 403.29 a | 902 ± 290.19 b |
| PAR | SI | 158 ± 39.59 b | 259 ± 33.88 ac | 193 ± 93.20 bc | 131 ± 28.22 bd |
| | SII | 107 ± 17.04 a | 106 ± 30.54 a | 99 ± 30.13 a | 65 ± 22.58 b |
| | SIII | 70 ± 7.53 b | 57 ± 14.92 b | 70 ± 16.47 b | 141 ± 40.69 a |
| | SIV | 74 ± 24.52 b | 112 ± 36.62 b | 100 ± 66.12 b | 302 ± 58.66 a |
| Y(II) | SI | 0.677 ± 0.071 a | 0.591 ± 0.20 ac | 0.693 ± 0.07 ab | 0.712 ± 0.03 ab |
| | SII | 0.578 ± 0.11 b | 0.659 ± 0.06 ad | 0.611 ± 0.09 bd | 0.728 ± 0.73 ac |
| | SIII | 0.685 ± 0.04 a | 0.713 ± 0.03 ab | 0.675 ± 0.04 a | 0.667 ± 0.07 ac |
| | SIV | 0.693 ± 0.08 a | 0.662 ± 0.05 a | 0.683 ± 0.07 a | 0.646 ± 0.09 a |
| ETR | SI | 44.49 ± 11.65 b | 65.87 ± 40.53 a | 55.78 ± 12.26 b | 39.34 ± 8.79 b |
| | SII | 26.18 ± 7.70 a | 29.73 ± 9.61 ab | 25.00 ± 7.54 a | 19.88 ± 7.13 ac |
| | SIII | 20.16 ± 2.99 b | 17.06 ± 4.42 bd | 19.81 ± 5.24 bd | 39.41 ± 11.57 ac |
| | SIV | 21.01 ± 6.43 b | 30.61 ± 8.41 bd | 27.54 ± 7.51 bd | 81.62 ± 17.05 ac |
| SPAD | SI | 28.53 ± 3.58 b | 29.73 ± 3.45 bd | 32.46 ± 2.48 ad | 32.87 ± 3.61 ac |
| | SII | 18.6 ± 2.14 a | 20.13 ± 1.85 a | 19.97 ± 3.87 a | 20.61 ± 3.49 a |
| | SIII | 15.58 ± 4.35 b | 18.24 ± 5.37 b | 16.02 ± 2.99 b | 20.47 ± 3.49 a |
| | SIV | 17.90 ± 5.28 a | 16.72 ± 3.84 a | 17.53 ± 5.82 a | 19.40 ± 1.69 a |

[1] SD; C—control; L—laser light; $p$—magnetic field; AgNC—silver nanocolloid; SI—budding, SII—onset of flowering, SIII—flowering, SIV—fructification. Different letters a→d—significant differences ($p \leq 0.05$), the same letters—no difference between the control and study groups.

The mean value of Y(II) was within the range from 0.644 (SII) to 0.685 (SIII) and the ETR value from 24.11 (SIII) to 51.37 (SI) (Figure 6). In the present study, the application of silver nanocolloid produced an improvement in terms of photochemical efficiency, particularly at the stages of: budding and onset of flowering.

Based on the data presented in Table 5, no significant correlation between the Y(II) and ETR at the respective stages can be identified.

Average correlation (significance at the level of $\alpha = 0.05$) could only be observed between the SII and SIII (r = 0.408) and between the SII and SIV (r = 0.465).

To recapitulate the presented study, it can be observed that the use of various techniques intended as a way to improve germination as well as photosynthetic pigment content and photosynthetic efficiency, yielded varying results. The greenness index at the stages of: budding, onset of flowering, and flowering was higher in all plants grown from seeds treated with laser light and alternating magnetic fields as well as soaked in silver nanocolloid, when compared to the control. On the other hand, the same was lower at the stage of fructification due to the aging of leaves and their change to a more yellowish-orange color.

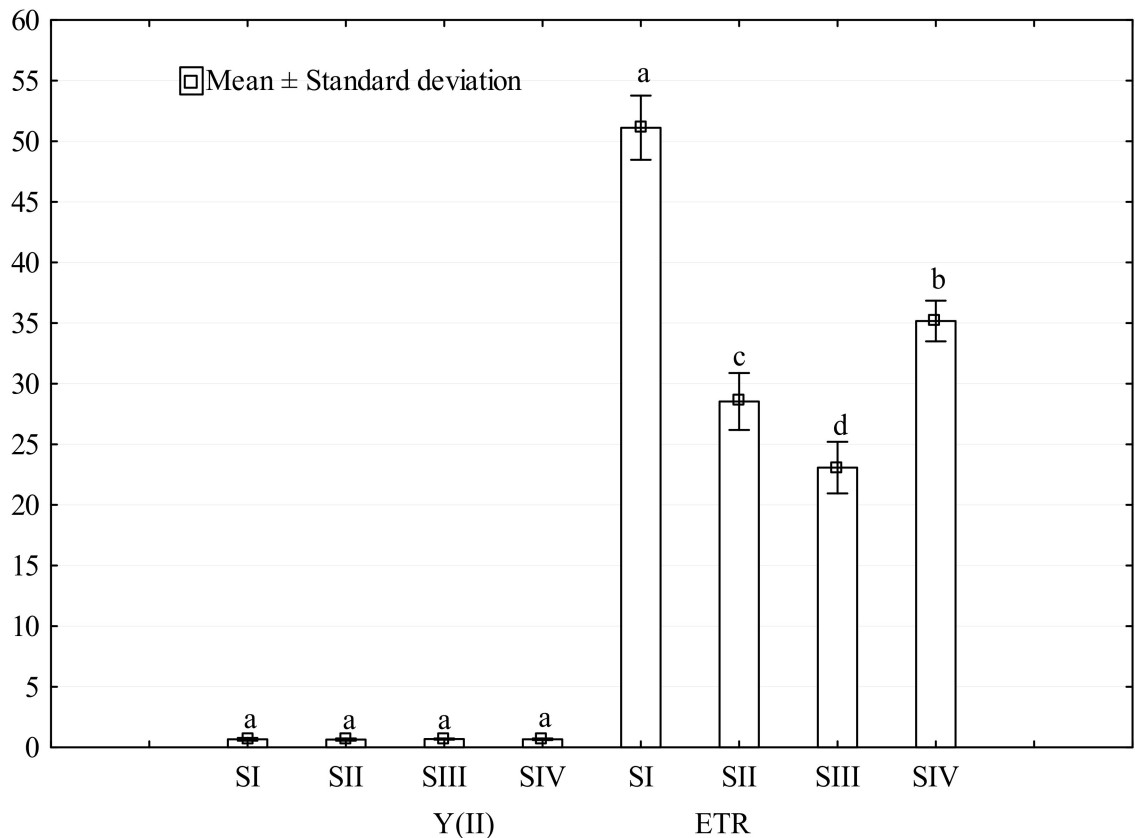

**Figure 6.** Mean values of photochemical efficiency and electron transport at the respective developmental stages. Different letters a→d—significant differences ($p \leq 0.05$), the same letters—no difference between the control and study groups.

**Table 5.** Pearson r correlation coefficient between Y(II) and ETR at the respective stages of plant development.

| Stages of Plant Development | Pearson r Coefficient | | | |
|---|---|---|---|---|
| | SI | SII | SIII | SIV |
| SI | 0.218 | −0.139 | 0.108 | 0.085 |
| SII | 0.006 | 0.268 | 0.408 | 0.465 |
| SIII | 0.180 | 0.143 | −0.022 | −0.052 |
| SIV | −0.074 | 0.113 | −0.185 | −0.235 |

SI—budding, SII—onset of flowering, SIII—flowering, SIV—fructification.

## 4. Conclusions

The best effects in terms of increase of the germination energy and capacity were observed in the case of seeds stimulated with magnetic fields, where the same improved, respectively, by 20% and 4% relative to the control. The emergence and density of plants deteriorated in all study samples. At the stage of the onset of flowering pumpkin, an improvement in terms of photosynthetic efficiency and greenness index was observed in all study groups. The highest improvement was recorded for seeds soaked in nanocolloid of silver and reached 23% (intensity of photosynthesis) and 11% (greenness index).

The highest chlorophyll *a*, *b* and carotenoid content was recorded in leaves grown on Petri dishes from seeds soaked in nanocolloid of silver. The increase relative to the control was, respectively, 53, 11, and 79%. In turn, leaves of pumpkin grown in pots displayed the highest content of chlorophyll *a*—42 and 43% increase (*p* and AgNC), chlorophyll *b* by 25% (*p*), and carotenoids by 66 and 81% (L and *p*) as compared to the control.

**Author Contributions:** A.D.-H., M.G. Conceptualization, Methodology, Formal Analysis, Data Curation, Writing—Original Draft; M.K., M.G., M.S. Investigation, Resources, Writing—Original Draft. All authors have read and agreed to the published version of the manuscript.

**Funding:** This research received no external funding.

**Data Availability Statement:** Data contained within the article.

**Conflicts of Interest:** The authors declare no conflict of interest.

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
