# Peer review of "Influence of Silver Nanoparticles, Laser Light and Electromagnetic Stimulation of Seeds on Germination Rate and Photosynthetic Parameters in Pumpkin (Cucurbita pepo L.) Leaves"

_applsci, doi:10.3390/app11062780_

Round 1

Reviewer 1 Report

The file should have the line number so the reviewer can mention the line number for each point:

Title: Please use upper case for each highlighted word

Line 13 Abstract: please edit, it should be "and"

Line 4 of Introduction: approx. it is not formally correct so please use "approximately"

age 2 Line 12: Please edit highlighted section as follows:

Nanoparticles are now being used in the manufacture household goods, food packaging, feed production, as well as textile and cosmetic products.

Page 2 Line 21 and 22:

[26,27],

[27,30] (please delete the space before 30)

Page 2 Line 28: seed germination and seedling establishment.

Page 2 Line 30: the seeds [35]. please delete the material

All subheadings should be in italic and has number (follow the journal format)

Page 2 Line 41: Materials and Methods

Page 2: Line 42: Maybe could elaborate a bit further and write the name of the company and some information on seed package such as total germination or 100 seed weight (why you choose this seedlot, to begin with it has a poor germination and with your techniques you were able to enhance the emergence rate?

Page 3 line 23: Please rewrite this section perhaps it should be: "on laboratory Petri dishes and in pots in the greenhouse"

Line 29: May you please double check your numbers, are you sure that 3 layers for blotter paper can be moistened throughly with 2 ml DI water?

Page 4: Please use ( ) instead of [ ]

Page 5: statistical analysis: Please explain more about any data transformation of germination % and the normality test that you may done before subject the data to ANOVA 

Page 7 Figure 3 b): to be consistence please add line around each bar same as fig 3 a) Also please add to the figure caption: what are those error bars; SD or SE (They should be SD since in Fig 4 and 5 the SD is in the Figure legend

Page 13-15: All references should be reformat and follow the MDPI journal's instruction for references preparation. 

Reviewer 2 Report

From our point of view, the work has the novelty of using AFM microscopy to estimate the height, histogram and distribution of the nanoparticles of 100 objects.

However, the reader needs to know precisely

(i) The detailed procedure for preparing the nanocolloids and the concentrations tested.

(ii) Know by SEM-TEM and / or AFM the shape of AgNP nanoparticles distributed by [System Poland] and the resulting AgNC and discuss the role and shape and size of AgNC on possible cell damage.

(iii) Reflect in detail the measurement procedure and the number of determinations carried out with the AgNC group, and the L and P groups. The concentrations, times and stages and all the results should be offered in a Table: heights and germination rates with the AFM analysis equipment and by conventional methods.

 In the text it is said that the AFM preparation was obtained by placing a drop of the solution (of what concentration?) On a mica plate attached to a metal disk and evaporating it. The measurements are also said to have been made at room temperature and AFM images were processed using nanoscope analysis software [Bruker, USA]. The procedure for determining that average surface area of ​​16414.2 nm2, which represents 75 AgNC, and the determination of the heights in nm, is required to be explained in more detail.

(iv) Compare and offer the determinations of the heights and germination indices by the conventional methods, against the AFM methods and compare the results

(v) It is necessary to compare the results discussion, the advantages of using AFM technologies, compared to conventional ones. We think it would be convenient to include a summary table

(vi) Missing in the Tables made specify the acronyms used, at least at the bottom of the figure

Reviewer 3 Report

The introduction part - lack of aims of research, materials, methods, results - poorly described, lack of conclusions and discussion - only presents literature data without conclusions. The manuscript is written messily.

Round 2

Reviewer 1 Report

The authors rewrote and edited several sections of the resubmitted draft and significantly increased the quality of the content of the manuscript. 

I have some minor editorials and some errors in citations that need to be edited carefully and elaborately to enhance the quality of the presentation and scientific soundness of this valuable work. 

Page 1: A suggestion to be consistence either use 53 and 11% ...42 and 43% or 53% and 11% ... 42% and 43%

The abstract should be in 1 paragraph. So please check the journal requirements and edit accordingly. 

Please use a capital letter for each word on the heading and sub-headings throughout the manuscript  

Page 5: 3. Results and Discussion

Page 7: Should be by Gubbins et al. [65] 

Stampoulis et al. [68]

Abdel-Azeem and Elsayed [69]

Please double-check the references list: There are some confusion and errors in the numbering of references, please edit:

Barren et al. [70]

Yasure and rani [71]

Mehrian et al. [72]

Page 9: Liu et al. [73]

Page 9-Line 3: Please edit this error there is no Liu et al in references that conducted a study on cucumber 

Kachel et al. [74]

Please double-check all references and match them with the cited one in the body of your manuscript. 

Reviewer 2 Report

The work has been improved with the corrections introduced, but nevertheless we observe that before accepting the article, a more complete Table must be offered than the one provided (Table 2), and that shows all the results, and any reader can verify that the increases or decreases with respect to the control group are those specified by the authors:
 "The conducted analysis showed that an increase was observed in relation to the controls GE by about 11% for L, by about 37% for P, and a decrease by about 1.8% for AgNC, and by GC of about 5.9%, 4.7% and 10.7%, respectively, for Ne8, a decrease was observed for all values ​​tested, by about 1.6%, 50% and 83% respectively for L, P and AgNC. There was no change from L for Ne14, and for P and AgNC there was a decrease of approximately 50% and 66%, respectively "

We also find some aspects that must be corrected and that we indicate:
1. Introduction: See error in "f" "near reference [46], where it says:" f Boswellia ovaliofoliolata trees [46] "
2.3. The experiment of pumpkin germination: Experimental data with regard to the time-frames are provided in Table 1. The heading of Table 1 is not indicated. The BBCH scale is not explained in methodology and how the increments can be deduced from this table, decreases with respect to the control group, and that are not reflected in Table 2.

Reviewer 3 Report

The seeds were stimulated prior to sowing using the following methods: He-Ne laser light (Fig. 1) at the wavelength of 632.8 nm; surface power density of 6 mW∙cm-2, with the expo-sure time of 1 minute (L); alternating magnetic field with 30 mT induction with the expo-sure time of 1 minute (P); silver nanoparticles (AgNPs) – by way of soaking the seeds in silver nanocolloid (AgNC) and non-stimulated samples C (control). Figure 1 shows a de-vice for the stimulation of He-Ne laser light.

Experimental data with regard to the time-frames are provided in Table 1.  It should be  Table 1. Experimental data with regard to the time frames.

The BBCH-scale – add a piece of information about this scale in the text (Material and Methods)

Numerous double spaces.

Lack of dots.

One big gap at the end of the abstract (what for?).

Many short paragraphs with only one sentence are in the introduction. Please combine these sentences in one paragraph

Different sizes and fonts are used in the manuscript.

in situ - italic 

Pmol or pmol ?

Two 3.1 paragraphs. Please change the number of the second one.

In 3.1 and next paragraphs will be nice to see some short introduction to performed analysis.

Different letters a→c – significant differences (p ≤ 0.05), the same letters – no differ-ence between the control and study groups. – The sentence is related with Table 2, so should be near the rest of the description. Now it seems that is in the main text.

The conducted analysis showed that an increase was observed in relation to the con-trols

GE by about 11% for L, by about 37% for P, and a decrease by about 1.8% for AgNC, and by GC of about 5.9%, 4.7% and 10.7%, respectively  - for AgNC and for GC there is no statistical difference. The only statistical difference is important.   

magnetic feld?

crude fber?

fnal yield ?

C -control; L - laser light; P – magnetic field; AgNC - silver nanocolloid

Figure 3. Content of photosynthetic pigments in the leaves of pumpkin plants grown from seeds subjected to the respective pre-sowing treatments: a) leaves from dishes, b) leaves from pots.

Different letters a→d – significant differences (p ≤ 0.05), the same letters – no differ-ences. Statistical analysis performed separately for Chlorophyll a, b, and carotenoids

I think  that is better (below)

Figure 3. Content of photosynthetic pigments in the leaves of pumpkin plants grown from seeds subjected to the respective pre-sowing treatments: a) leaves from dishes, b) leaves from pots. C -control; L - laser light; P – magnetic field; AgNC - silver nanocolloid. Different letters a→d – significant differences (p ≤ 0.05), the same letters – no differences. Statistical analysis performed separately for Chlorophyll a, b, and carotenoids.

I think that Table 3 and 4 could be combined in one table.

C –control; L – laser light; P – magnetic field; AgNC - silver nanocolloid

Fig. 4. Mean content of chlorophyll a+b and carotenoids (in seed leaves grown in dishes for 8 days) . Dif-ferent letters a→b – significant differences (p ≤ 0.05), the same letters – no difference between the control.

I think  that is better (below)

Fig. 4. Mean content of chlorophyll a+b and carotenoids (in seed leaves grown in dishes for 8 days) . Dif-ferent letters a→b – significant differences (p ≤ 0.05), the same letters – no difference between the control. C –control; L – laser light; P – magnetic field; AgNC - silver nanocolloid

The same for Fig 5.

Figures 4 and 5 present the mean values of chlorophyll a+b and carotenoid content in seed leafs grown in dishes and pots. Calculations of the correlation coefficients between the values of total chlorophyll (Fig 4, 5) and carotenoids revealed no correlation between the Petri dish study groups. –Fig. 4 and 5 there are only mean content of chlorophyll and what is with coefficients?

Presented in Table 5 are the results of chlorophyll fluorescence measurements in the leaves of pumpkin grown from seeds subjected to various pre-sowing treatment. PAR radiation was low (Table 5), ranging from 57 (L, SIII) to 302 μmol of photons m-2s-1 (AgNC, SIV).

The description under table 5 should be without the gap. 

1)SD; C –control; L – laser light; P – magnetic field; AgNC - silver nanocolloid; SI - budding, SII - onset of flowering, SIII - flowering, SIV - fructification. Different letters a→d – significant differences (p ≤ 0.05), the same letters – no difference between the control and study groups.

Carthamus tinctorius - italic
